# Mature and Older Adults’ Perception of Active Ageing and the Need for Supporting Services: Insights from a Qualitative Study

**DOI:** 10.3390/ijerph19137660

**Published:** 2022-06-23

**Authors:** Valentina Barbaccia, Laura Bravi, Federica Murmura, Elisabetta Savelli, Elena Viganò

**Affiliations:** Department of Economics, Society, Politics, University of Urbino Carlo Bo, Via Saffi 42, 61029 Urbino, Italy; v.barbaccia@campus.uniurb.it (V.B.); laura.bravi@uniurb.it (L.B.); elisabetta.savelli@uniurb.it (E.S.); elena.vigano@uniurb.it (E.V.)

**Keywords:** active ageing, life-course perspective, older adults’ lifestyle, adult population, healthy behavior, socio-relational behavior

## Abstract

The improvement in life expectancy, economic conditions, and technological and medical progress have radically changed the demographic structure of many societies. Since many countries now have an ageing population, by adopting a life-course study perspective, this paper aims to explore the needs of older adults (over 60), and the currently adult population which will become older in the coming decades (50–60 years). In detail, the study investigates the lifestyles of the target populations by focusing on two main areas concerning health (healthy diet; attitudes towards physical activity) and socio-relational-housing and living conditions (social housing, senior co-housing in rural environments, etc.). A qualitative study was carried out based on 16 in-depth interviews developed over one month (February 2022). The conduct of the interviews was supported by the Italian Center for Sensory Analysis (CIAS). Emerging from the results, the concept of active ageing is perceived by mature and older adults in a positive and optimistic way. The sample considered want to re-engage in life, continuing to be active, useful, and maintaining their self-esteem, social life and independence. However, despite older people’s major concerns being preserve their physical abilities and social integration, this target group adopts behaviours focused more on current well-being rather than worrying too much about how this well-being will change as they age.

## 1. Introduction

Older adults now represent a growing sector of the population since they are living longer, and the world’s population is increasingly ageing. In 2019, older adults were 703 million worldwide and the World Health Organization (WHO) expects that, in 2050, this number will increase to 1.5 billion [1]. 

Increasing life-expectancy, together with the relatively low age of retirement, raise important questions regarding the role of older people as individuals and their role in society.

Ageing and its related behaviours are multi-dimensional constructs [2]. According to Nunan and Di Domenico [3], older adults’ lifestyles are affected by four different disciplinary perspectives on ageing: (i) the biological/physical perspective, which involves the body’s variations affecting physical health and functionality, such as decline in vision and hearing [4]; (ii) the psychological/cognitive perspective, including changes which occur in personality, cognitive ability and self-concept throughout ageing; (iii) the social perspective, involving relationships and the evolution of roles and responsibilities across the generations; (iv) the environmental/contextual perspective, linked to the physical circumstances and collective experiences which may affect the consumers’ mindset, while triggering specific needs. 

Although ageing of the population occurs for many reasons, non-genetic factors tend to have a much greater impact on ageing than genetic ones [5]. Indeed, external variables linked to adult lifestyle—such as food and alcohol intake, smoking, and social and economic conditions—can determine important diseases and health issues, such as hypertension, cancer, and diabetes. However, this process, when caused by external factors, may be reversible at any age [6]. For example, maintaining healthy behaviours throughout life—such as healthy habits, balanced diets and regular physical activity—contributes to reducing the risk of diseases, improving physical and mental capacity and delaying care dependency [7,8]. On the other hand, social exclusion and low socioeconomic status can have a negative influence on older adults’ cognition and health. For example, financial issues caused by low income can raise stress and increase blood pressure and depression [9]. 

The emergence of the concept of active ageing, which belongs to the area of social policy, can be seen as part of a wider change in the way the meanings of ageing are constructed and how the position of old age in an individual biography is interpreted [10]. Active ageing can be intended as “the process of optimizing opportunities for health, participation and security in order to enhance quality of life as people age”. In this respect, ‘active’ means “continuing participation in social, economic, cultural, spiritual and civic affairs, and not just the ability to be physically active or to participate in the labour force” [11] (p. 12). Based on a “comprehensive approach” to ageing, Foster and Walker (2021) argue that active ageing should be widely preventative and inclusive, involving people’s aging throughout their entire lives, and including them whatever may be their physical, psychological and social condition. More specifically, according to the WHO [6], a good lifestyle in an active ageing perspective concerns the interaction between the individual and his/her surroundings, involving biological, psychological and social domains. Thus, it should start early in life, including participation in family and community life, healthy and balanced diet, adequate physical activity, avoidance of smoking and excessive alcohol consumption, and active ageing policies should promote the implementation of preventative health treatments to enhance lifestyle, food, and consumption habits at all stages and life situations [5]. To do so, alterations in preferences and constraints that may emerge throughout adulthood must be evaluated [7] to provide effective actions and measures aimed at improving the overall quality of later life.

Although this topic is increasingly discussed and included in the political agenda of public institutions, governments, and private organizations, literature on active ageing is still lacking in contributions investigating the consumer perspective. Except for a few studies, such as that of Tešin et al. [12] focused on cultural events, and that of Ebekozien [13] analysing home ownership for senior citizens, most of the existing research has adopted a conceptual approach (e.g., Lassen and Moreira [14]) or investigated the supply service perspective (e.g., Lassen [15]), ignoring in-depth explorations of older adults’ needs and preferences.

However, such an investigation could be critical in adapting service content and understanding which services are most compatible with the current expectations of this market segment under an active ageing perspective.

By adopting the life course study perspective [16,17], this paper aims to explore the needs of older people (over 60) focusing on two main areas concerning health (healthy diet; attitudes towards physical activity) and socio-relational-housing living conditions (social housing, senior co-housing in rural environments, etc.), which the WHO [18] recognizes as the main domains of active ageing. Despite the focus on older adults, this study also includes in its analysis the currently mature population (50–60 years)—which will become older in the coming decades—and its progressive ageing to provide a deeper understanding of the overall lifestyle of adults and their current and future active ageing expectations. 

Two main research questions operationalize the above objective:⋅RQ1: How do mature and older adults perceive the concept of ‘active ageing’? What are their current habits and expectations regarding health and social conditions? ⋅RQ2: Which kind of health and social services are mainly requested and could be improved to support the population towards active ageing?

This paper will contribute to the existing literature on active ageing, by deepening the consumer perspective that has been neglected by previous research and by making a comparison between two consumers’ age groups, i.e., mature and older adults, covering an existing gap in the reference literature, so as to have a term of comparison between the age group directly involved in these practices and the one that will be involved in a few years. It uses a qualitative methodology based on semi-structured interviews to gain insight and understanding of older people’s perceptions, preferences, physical situation, current habits and expectations regarding health and social conditions.

The paper is structured as follows: Section 2 describes the literature background on the concept and domains of active ageing with a focus on health and socio-relational dimensions regarding this issue. Section 3 describes the methodology used; Section 4 presents the results obtained from interviews. Then Section 5 discusses the results and related implications, while Section 6 draws the main conclusions.

## 2. Literature Background

### 2.1. Older Adults’ Needs and Preferences: A Life Course Perspective

Social, behavioural and biomedical scientists have long promoted an interdisciplinary Life Course Perspective (LCP) for organizing research on human development, maturation and ageing [16,19]. The LCP has been used by researchers across several disciplines and, although its definition may vary depending on the investigation’s background, its most common use refers to stability and change [20]. According to Moschis [21] (p. 2036), “life course theoretical perspectives explain stability and change of thoughts and actions as a result of a person’s adaptation to changing life events and conditions experienced earlier in life, with the onus of explanation on mechanisms or adaptation processes that bring about changes”. Life events lead to various physical, social, and emotional demands to which a person must adapt, with results in a changing construct of thought, behaviours and needs [22]. In this respect, ageing can evolve in different directions and its understanding is a key factor to reduce societal and individual costs for the older population. In other words, the life course model considers behaviour at any life stage, supposing a relation of interdependence between earlier-in-life experiences and later-in-life consequences. 

By assuming this perspective, it is clear how important it is to investigate the main influencing factors on ageing and the older customers’ view of this phenomenon. As suggested by Weaver et al. [22] (p. 247) and other consumer researchers, “studying of prior events in individuals’ lives as well as their perception of the future can help to analyse and understand different patterns of consumer behaviour across various market”. Thus, it is increasingly essential to understand older adults’ needs and preferences throughout life, and not just because the older population is growing, but because they also contribute in many ways to the economy and society [23]. 

Older adults are an active consumer group and, thanks to their experience, they take attentive purchasing decisions. As observed by Rousseau and Venter [24], this cohort has specific needs and preferences for products and services. For example, they need easy-to-open packaging and easy-to-read labels, especially when purchasing nutrition and health products or dining at restaurants. Reading barriers should be removed in restaurants. Dixon [25] found that only few older clients can read a restaurant menu without glasses as writing features small characters. This offers a wide space for reflection: a focus on older adults should be strengthened even in such common and simple contexts. Dining experiences are very valuable for older adults, mainly for two reasons: dietary habits are crucial in maintaining proper health, and experiences of eating out have a positive impact on quality of life and socialization. 

Eating out, being physically active, visiting shopping malls, traveling, participating in recreational activities, reading, going to movie theatres or other entertainment venues, are the ways in which older adults mainly enjoy spending their leisure time [24,26,27].

Having more time than younger generations, older adults invest time in searching for good value for money, shopping for convenience and sharing information with others. Lambert-Pandraud et al. [28] observed that older adults tend to purchase brands that they have known well for a long time, probably due to previous experiences, attachment to such products, habits, nostalgia or maybe aversion to change.

As argued by Rosseau [26], older adults are also sensitive to services and facilities they receive from businesses such as shops, hospitals, financial and government organizations. For example, outdoor mobility services are key-factors in maintaining older people’s independence. To improve people’s experiences and achievement of mobility, transport services need to be perceived as secure and friendly, inspiring confidence and comfort [29].

Overall, older adults of today have a more active lifestyle and higher expectations than past cohorts. People over 50 also feel and want to look younger than their chronological age and they do not want to be stereotyped as ‘old’ [30,31]. However, the ageing course evolves in a different way from person to person and people’s lifestyles between their 50s and 60s may change within different scenarios. For instance, although some adults are parenting and working full-time, others may be enjoying their retirement and may be grandparents [32]. Different contexts also lead to dissimilar needs, lifestyles, and consumption habits and their implications must be considered in promoting well-being and an active ageing [11,31].

### 2.2. Active Ageing: Concept and Domains

The concept of ‘active ageing’ developed in the 1990s and it usually refers to individual or collective strategies for optimising economic, social and cultural participation throughout the life course [5,33,34]. Active ageing emphasizes the relationship between health and activity [35] and it usually falls into the concept of healthy ageing, based on the development and maintenance of functional ability that encourages wellness in later life [36,37]. Functional ability, indeed, allows older people to be capable of meeting their basic needs, learning, making decisions for themselves, having strong relationships, mobilising, and contributing to society [38]. Therefore, active and healthy ageing means getting older in good health, feeling actively involved in social activities, being independent in daily activities and feeling fulfilled in jobs and in social engagements. This perspective refuses to consider older adults as passive and non-independent, while emphasizing their autonomy and active role in society.

As older age tends to be stereotyped as a time of inactivity, ‘rolelessness’ and dependency [10], a development of attitudes and a more positive approach towards the ageing course are needed. On the face of it, an important role is played by governments and national or local strategies [39]. Making the world more age-friendly will not only improve customer satisfaction but can also break down barriers to overall healthy ageing and stereotypes or prejudices attributed to the older generation [40]. 

According to the WHO [41] (p. 1), “age-friendly environments aim to encourage active and healthy ageing by optimizing health, stimulating inclusion and enabling well-being in older age”, based on appropriate physical, social and municipal services.

In Europe, the extent of and progress towards active ageing are monitored by the Active Ageing Index (AAI), a policy tool—in use since 2012—which “assesses the untapped potential among older people across multiple dimensions of active and healthy ageing” [39] (p. 3). In doing so, the AAI helps policy makers to evaluate and understand the areas that need to be improved, to promote a balanced active ageing process. Moreover, it provides a further framework for active ageing domains on which research should focus its attention.

In more detail, this index measures the “level to which older people live independent lives, participate in paid employment and social activities as well as their capacity to actively age” (https://unece.org/, accessed on 22 February 2022). The AAI is composed of 22 individual indicators, grouped into four domain areas: (i) employment rate; (ii) social participation such as voluntary work, political participation and care for children, grandchildren and older adults; (iii) independent living which includes physical exercise, access to health services, financial security, physical safety and lifelong learning; and (iv) capacity for active ageing in relation to life expectancy, mental well-being, access to information and communication technologies (ICT), social connection and level of education. Specifically, employment, social participation, and independent living measure achievements towards active ageing, while the fourth domain offers a view of preparedness for achieving positive results [42].

Overall, the literature on active ageing highlights the importance of focusing on the health and socio-relational domains of ageing. According to Tavares et al. [43], the understanding of healthy aging is relevant for older people, even if affected by diseases. In this context, there is the need to develop the skills and the environments that make older adults able to age in the best possible way. Among these aspects, good nutrition, physical activity and cognitive health are key factors for living a high-quality late life. 

However, the maintenance of good health is also influenced by the socio-relational context. Some researchers found that formal or informal care of ageing people is a critical need [44,45]. Older people’s care should be encouraged by different perspectives, such as health and social care, transportation, housing and urban planning, always promoting the socialization and the inclusion of older adults. 

Thus, it is evident that the quality of this segment is strictly connected to physical, cognitive health and overall care, which represent the basic needs of older adults. As a result, to promote active ageing, the way in which societies are structured in all sectors should be adapted to older adults’ needs.

### 2.3. The Healthy Domain of Active Ageing

According to the Heikkinen and WHO [46], the concept of health is formed by the interactivity between the individual and his/her physical and cognitive capacities, and environmental characteristics such as home, community, and social surroundings. Hence, health is a functional status, which can be defined as “a person’s ability to perform the activities necessary to ensure well-being. It is often conceptualized as the integration of three domains of function: biological, psychological (cognitive and affective) and social” [46] (p. 2).

The WHO’s perspective on healthy ageing also relies on two important factors: diversity and inequity. An example of diversity is provided by Rudnicka et al. [47], who affirm that sometimes the physical and mental capacities of an 80-year-old person can be like those of a 30-year-old person, while sometimes people may perceive a decrease in capabilities much earlier in life. Diversity issues may arise from inequity, described by Venkatapuram et al. [48] as the influencing factors on life course such as characteristics related to genetics, sex, ethnicity, and environment. These inequities should be reduced through the implementation of an effective programme for healthy ageing, together with a clear understanding of customers’ needs and an approach to people most at risk of poor health and exclusion [41].

According to Vintilă et al. [49], innovative health promotion initiatives are needed. These strategies should stimulate social processes and promote the concept of health as an integral part of the life course. People should be encouraged to believe that they can live healthily, of managing their difficulties, and of taking responsibility for their own well-being. 

The extent of older adults’ contribution to society depends heavily on the maintenance of good health. In this respect, the WHO has proposed concrete priorities to undertake a “Decade of Healthy Ageing (2020–2030)”. Specifically, it promotes four key actions to be taken in these years: adjusting the way of thinking, feeling, and acting towards age and ageing; developing communities to encourage abilities in late life; delivering integrated care and primary health services that are responsive to the needs of older people; and supporting older people needing access to long-term care [50].

Evidence shows that ageing may be related to the arising of chronic diseases, physical disabilities, mental illnesses and other co-morbidities [51] and the factors affecting these medical problems are several. Among them are nutritional, physical and psycho-emotional concerns, limited awareness of risk factors, health-care systems, and social concerns. As observed by Halaweh et al. [52], older adults’ major concern is for their cognitive health and a possible decline in memory. However, good cognitive health is influenced by social environments, independence, and life activities, and it may be maintained and improved through a mentally and physically active lifestyle and a positive attitude towards aging [53,54,55].

Preserving good physical health and functioning is critical to facilitate older adults’ mobility and daily activities. For example, this also decreases the occurrence of accidental falls [56,57]. Results from a study by Fastame [58] on the Italian context showed that older adults who are more physically active reported a better cognitive condition and a higher satisfaction in life than sedentary participants did. Yet, McPhee et al. [59] believe that older adults’ participation in physical activities has still largely to grow, especially amongst people living in less affluent areas. According to Lockenhoff and Carstensen [60], people getting older tend to be more interested in maintaining their current abilities rather than improving their health. Sport participation is often seen as a form of leisure time able to provide a sense of belonging and an increasing socialization [61]. Thus, other than being encouraged by clinicians, families, or friends [59], the WHO [6] suggests that both individual and policy actions should be implemented through the life course. For example, while people should engage themselves in exercise activities from early life to older age (even in a light form such as walking, climbing stairs or doing housework), policy makers could “incorporate exercise into school curricula”, “create workplaces which provide exercise facilities” and “encourage sports for seniors” [6] (p. 21).

Together with exercise, good nutrition has a crucial role in preserving active ageing and a high-quality of life [62]. Different studies have shown that healthy diets positively affect physical activity, condition, and mental wellbeing. However, Rusu et al. [63] found that malnutrition is becoming far too common among the fast-growing older population. Their research and that of Ganesan et al. [23] revealed that most older adults are not able to eat entire portions and that during ageing they have decreased food intake. Thus, this has led to a reduced nutrient intake. Additionally, they do not consider the purchase of functional food, due to a lack of nutritional knowledge [64]. According to Vintilă et al. [49], consumer behaviour is influenced by the quality and the amount of health information available. Older customers may be confused by conflicting information, encouraged by useful information, or deceived by fake or insufficient news. Thereby, customers need fundamental abilities as well as a strong support network to assist them in perceiving the correct information in the appropriate manner. Active ageing policies should strengthen older adults’ education regarding eating behaviour—for example, through more medical advice increasing consumer awareness of the close connection of good nutrition with health [6,49]. On the other hand, the individual’s target is to “maintain a normal body weight” along the life course and usually prefer a “diet high in fibre and low in animal fat and salt” [6] (p. 21). As older people’s eating behaviour is strongly influenced by social context, Rusu et al. [63] also suggest that eating in groups may be a good motivational strategy for having healthy meals.

Given these critical contexts, various research (e.g., Shen and Tanui [65]) is based on assuming political/institutional prospects aimed at improving the old-age social welfare system. From this perspective, it is common to incentivize a medical approach based on the study of diseases or ageing issues. However, the literature should be widened by studies that investigate the older customer’s perspectives. In fact, except for a small amount of research, previous studies neglect the importance of in-depth explorations of older adults’ needs. To develop correct and effective service content, it is necessary to understand which are the basic preferences that older people need to be satisfied and which are their real expectations regarding their current and future health condition. According to the study conducted by Silva and Fixina [66], older adults accept old age and their current physical condition. They consider the ageing process as a life phase in the middle between a ‘state of health’ and a ‘state of illnesses’. Although they are comfortable in being in their last phase of the life cycle, older people do not have a vision of the future and perceive a negative feeling in thinking about it. Given these premises, this paper seeks to contribute to the existing active ageing literature, investigating barriers and triggers that could contribute toward the implementation of age specific health interventions and the diffusion of more positive expectations. 

### 2.4. The Socio-Relational Domain of Active Ageing

After good health, the maintenance of social relationships and engagement in collective leisure activities have an essential role in active and healthy ageing [49,67]. Increasing participation in social and recreational activities can enhance physical, cognitive, and emotional health and well-being in later life, with a consequence in reducing expenditure on health care provision [9,68,69]. Thus, the importance of neighbourhoods and communities in which people reside increases as they get older. People in age-friendly environments are empowered to live independently in good health, stay connected in their communities, and keep being socially integrated and active in a variety of roles, such as neighbours, friends, family members, colleagues, and volunteers [41].

However, ageing may hinder moments of socialization. Retirement, distance from families, and other physical issues decrease opportunities for meetings in person. Additionally, the COVID pandemic and lockdown measures had huge effects on the psychological well-being of older adults, increasing the threat of illness and loss of social support [70]. According to Jowell et al. [71] (p. 10), “the pandemic generated a sense of vulnerability in many older people who were previously enjoying a newfound sense of healthy ageing”. In this context, technologies, internet and social media have played a crucial role in providing alternatives to various daily activities, as well as for replacing physical encounters with virtual interactions [72], although the majority of older adults are not as proficient in using digital solutions as younger people are and this increased their exclusion and feeling of loneliness [69]. Thus, as suggested by Martins van Jaarsveld et al. [73] and Martínez-Alcalá [74], digital literacy among older adults is needed.

In the context of the pandemic, the content of leisure activities of older adults has also been greatly transformed and restricted. Participation in leisure activities allows them to improve mental health [75] and to re-engage with life once freed from middle-age responsibilities [76]. As argued by Rosseau [26], older adults find involvement in engaging themselves in community activities such as clubs, churches, non-governmental organizations, voluntary activities, or other social groups. Yet, in addition to receiving assistance, older adults also aim to assist others. For many, supporting people is the best way to boost their self-esteem and social involvement [43,77]. Furthermore, according to Silverstein and Bengtson [78], older adults have stronger religious beliefs than younger ones. This could be due to certain life course experiences, such as the losing of a partner, having financial difficulties or health problems. Being involved in a family environment, in communities, clubs or religious organisations, speaking out against ageism, and lifelong learning are actions which encourage older people to age actively [6]. 

Besides the importance of relationships and socialization, another critical aspect concerns housing conditions for older adults, which mainly depends on their health status. According to the WHO [6], the largest part of the population remains fit and able to care for themselves in later life. However, a minority of older adults is affected by health issues and age-related changes that lead them to the need for assistance in daily activities and sometimes institutionalization [79]. As has been widely pointed out, older adults prefer to continue living in their own house and neighbourhood. They often feel a strong attachment to their home, especially if they have lived in their house and community for most of their life [80]. 

Mattimore et al. [81] found that, in residential facilities, older people tend to feel lonely and marginalized. Additionally, as nursing homes often host both older adults with high levels of disability and people still with their full capabilities, activity management and entertainment activities may be difficult, leaving some residents without motivation [82,83]. 

Besides residential facilities solutions, the phenomenon of home care support through caregivers is increasingly common. However, the growing solution of integrating domiciliary care and similar support services may delay the need for residential care, but not entirely remove it [84]. 

Health is an essential requirement to allow older people to keep living at home, but it is not the only one. Other critical and influencing factors are financial status and family environment [85]. For example, Bookman [86] found that a large proportion of older adults has limited social contacts and feels isolated when ageing at home. According to Halaweh et al. [52], one of the most common concerns of this population group is avoiding becoming or being a burden to others. 

For all the above reasons, the WHO [41] states that interventions in the socio-relational domain of active ageing should be implemented by creating, maintaining and promoting supportive environments that enable social interaction and active lifestyles. To do so, there is the need to provide meaningful social activities that encourage older people to leave their homes and maintain supportive social networks. 

A variety of co-housing programs are emerging with the aim of providing solutions in between staying in place and an institutionalized stay. According to Rusinovic et al. [80], senior co-housing communities provide a middle ground for older adults who do not want to live in a nursing home but want the companionship of their peers. In fact, although co-housing residents live in separate flats, they spend time together sharing common areas, participating in social activities, and providing mutual support while respecting the boundaries. As previously argued by Bramford et al. [87], Rusinovic et al. [80] confirmed that people living in co-housing appreciate the social control and the safe residential environment provided by these communities. In addition, building relationships with residents decreases the feeling of loneliness and fears, which may characterize late life stages. The origin of senior co-housing lies in Denmark in 1987 [88] and it widely spread within Northern Europe thanks to appropriate public policies, legislation, and funding [89]. However, this model has been spreading in other areas only in recent years. Durante [90], observing the co-housing phenomenon in Italy, noted that its main contention point refers to affordability. Specifically, the author reflected that, when promoted as a private and bottom-up initiative, co-housing leads to a strong economic divide, which excludes lower- and middle-income people who find it hard accessing the housing market. 

A similar and suitable solution for this last class of people could be “social housing”. Hansson and Lundgren [91] (p. 162) define this model as “a system providing long-term housing to a group of households specified only by their limited financial resources, by means of a distribution system and subsidies”. The criteria of “limited financial resources” have to be defined in each particular country, on the basis of macro- and socio-economic conditions and available funds. Social housing, providing below-market rents or prices, is not self-supporting and needs public or private financial contribution.

In any case, in Italy the above housing solutions are rather uncommon. According to Sarlo et al. [92], older people are still mainly helped by family members and only marginally resort to institutionalisation and co-housing. However, in the Italian context, characterized by social barriers due to occupational precariousness or the dissolution of ‘traditional’ families, co-housing offers a solution which has created interest among the population. Despite this, living in co-housing is still widely seen as a utopian idea, a sign of the poor knowledge of this option [93]. For this reason, the areas of major intervention should be defined based on which facilities older adults expect and would like to have access to. To do so, the engagement and the identification of older people’s perspective is needed.

## 3. Methodology

To investigate older adults’ perception of active ageing, their expectations regarding health and social conditions (RQ1) and their specific needs for health and social services (RQ2), a qualitative study was carried out based on 16 in-depth interviews developed over one month (February 2022). 

The conduct of the interviews was supported by the Italian Centre of Sensory Analysis (CIAS), a service company belonging to Intertek Italia, specializing in qualitative research and sensory analysis of consumer markets.

The qualitative approach has been chosen as it makes it possible to gain insights about older adults’ perceptions, preferences, physical situation, current habits and expectations regarding health and social conditions. Notably, in depth-interviews are ideally suited to ‘create detailed pictures of people’s lives’ [94] (p. 229), and to understand the respondents’ interpretation of reality. Moreover, considering the peculiarities of the sample (in particular their age), in-depth and direct interviews are particularly suitable as a flexible tool that encourages spontaneity [95].

Each interview lasted around forty minutes and employed a semi-structured questionnaire based on open-ended questions grouped into the main topics, reflecting the list of domains used by the active ageing index [39,96]. Specifically, respondents were investigated on the following issues: (i) value profile (i.e., what are the main values orienting daily life and activities); (ii) knowledge and perception of active ageing; (i.e., what they intend for ‘active ageing’, how they represent it in their minds, which benefits are associated with this concept, which factors influence an ‘active ageing’ and what can society do to promote it); (iii) the role of sport and nutrition in their daily life (i.e., how much sporting activities and dietary patterns are considered to be important for active ageing and how much time respondents devote to sporting practice and food selection; which benefits they associate with sport and good eating patterns); (iv) perception and needs regarding sanitary assistance (i.e., how respondents perceive current sanitary assistance, and which kind of services they would like to see implemented); (v) knowledge and interest in different housing solutions (i.e., how they perceive social co-housing solutions); (vi) wishes for social, leisure and territorial service activities (i.e., which main services they appreciate and what they would like to be provided).

The respondents were selected among Intertek Italia’s community of panellists, after compiling a screening questionnaire to assess their electivity. Among all the eligible panellists, 16 participants were randomly selected and took part voluntarily in the research. Despite the lack of scientific method in determining the sample size of a qualitative study, our sample is in line with prior literature [97,98], suggesting that, for in-depth interviews, the sample size usually ranges between eight to ten. In a qualitative study, indeed, the researcher is not concerned about generalization from the sample to the population, as the emphasis is on information adequacy and richness [99].

Because people’s needs, lifestyles, and preferences tend to increasingly differ as they become older, the disparities between older adults, and their overall implications, must be considered [91]. Hence, the selection criteria for the sample have been set as follows:-100% residents in central Italy (i.e., ‘Area 3’ of Istat classification including Toscana, Marche, Umbria, Lazio), with a good regional mix;-50% residents in small/medium-sized towns (less than or equal to 50,000 inhabitants) and 50% living in larger population centres (over 50,000 inhabitants);-50% aged 50–60 (‘mature target’) [100] and 50% aged 61–75 (‘older adult target’) [101];-50% coastal residents and 50% inland residents;-50% women and 50% men;-good mix of family compositions (families of different dimensions; with and without children; and extended families);-good mix of educational levels and cultural interests.

The choice to make a comparison between mature and older adults, considering the two age groups as defined in the literature [100,101], was developed to have a term of comparison between the age group currently involved in these practices and the one that will be involved in a few years, allowing understanding of both points of view, thus developing current and more effective reference actions.

The moderator started the interviews by welcoming the participants and presenting herself. After illustrating the aim of the study and gaining permission for audio-recording of interviews, respondents were asked to briefly describe themselves and to rank their most important values in life. Then, the interviews proceeded by investigating the main research topics. 

Once completed, data were aggregated and analysed.

## 4. Results

### 4.1. Respondents’ Values and Their Overall Perception of Ageing

Analyzing data collected from interviews, a double perception of aging emerges: the current older adults or almost older adults stated that they feel younger and fitter than their predecessors, “I remember that when I was young, I looked at my father as if he was old at 45 years, but now I am 65 and I feel younger than him at 45”. In addition, both mature and older participants do not recognize themselves in the term ‘elderly’ with its negative connotations. Especially the older target showed a sense of pride and wellbeing in their age. They do not consider their stage as a ‘point of arrival’, but they perceive it as a point to be proud of, where people arrive thanks to all the important experiences made in life. Overall, participants are dynamic and active and they freely live their late-life once their work life had ceased, at least from the subjective point of view, “Ageing, the active one, is a way of life that leads you to do things that you never did before…”, “I am fully aware of my age and I accept it… each stage allows you to do things that could not be done in others” or “Aging is a continuous change that I perceive day to day… I learned to ask for favors, and I do it serenely”. This finding is consistent with Mathur and Moschis [30] and Moschis et al. [31], who argued that the older and mature adults of today not only have a more active lifestyle, but also do not want to feel old. However, whilst interviewees aged 61–75 accept being categorized as ‘older adults’, participants with age between 50 and 60 years perceived a feeling of inadequacy when being called ‘mature’ or ‘close to older stage’. The sample of respondents close to being considered as older adults endorse an optimistic view of later life: “Surely I will not stand in front of the window watching the weather”, “I will be able to take care of myself”, “I will do what I cannot do now for so many constraints imposed by life” or “I will certainly smile and I will finally be able to come back home, in Sicily, where I have all my loved ones”. Despite this perspective, societies tend to attribute a negative impact to ageing by associating this concept with decay, loss, inactivity and dependence [11]. 

Exploring the important values in the respondents’ life, this study allowed understanding of the reasons behind this positive perception of aging and the issues characterizing this life stage from a more active perspective. In order of importance, the values mentioned by participants are affection, respect, loyalty/honesty, faith, health, and personal and loved ones’ fulfillment. On this topic, no differences emerged between the two analyzed sample groups.

Specifically, the family situation is a segmenting factor for mature and older participants. In fact, senior grandparents support their children and grandchildren in daily activities, and they are often fundamental pillars in their lives. As stated by many older adults, this leads them to an active and rhythmic lifestyle: “We eat when our grandson needs to eat and, when we sit at the table, we adapt ourselves to his meals and snacks”, or “I take care of my granddaughter all afternoons, from when she came back from school until six o’clock when her parents pick her up”. On the other hand, people without family or with children who live far away perceive a greater loneliness and a relaxation in terms of routine: “I do not have much to do so I was here waiting for your phone call”, “I live alone, my daughter lives in London, I left my dogs with her because I was not able to take care of them”, “I talk a lot with my animals, they make me company” or “I am single, I do not particularly look for the company of others, I have neighbors but I have few contacts with them, I dedicate myself to my work and to volunteering”. 

During interviews, respect is considered as important because, as previously mentioned, society still tends to lead older people to feel the weight of ageing. In fact, older adults sometimes may feel discriminated against and abandoned: “When my brother-in-law was hospitalized, I felt very bad, I did not expect it and I did not expect that his family could do this with him”. During interviews, the topic of respect was often associated by older adults with family relationships, while the mature target group’s in-work activity made especial reference to situations experienced at work and out of the home. This aspect shows that even if a belief is shared, behind it daily routine has a great influence.

Loyalty and honesty are values mentioned by both participant groups in the broadest sense of awareness and responsibility towards others. Faith has a fundamental role for the older target group, and in many cases the mature group stated that their religious beliefs are increasing with the advancing of age. Moments of religious aggregation are also positively assessed from a social point of view (sociability and mutual help). 

Health is certainly a crucial value, even if, from the data collected, a poor level of prevention in the younger group of respondents emerged. Attention to their own health is rather greater in the older age group of respondents. For example, interviewees affirm that they control the age-related pathologies with which they are affected or with which they could be affected (e.g., osteoporosis, arthritis), through slighter daily movement and changes in nutrition. This result is related to the fact that older people recognize the importance in late life of taking care of health issues, while the mature target group consider prevention as a slow process, because from their perspective “there is still time to think about the decay due to ageing”.

Finally, work emerges as a fundamental variable and as a main factor in the dichotomy of activity vs. inactivity. Although ageing can be defined as a process, from the analysis of the interviews the main dividing line between the two groups of interviewees is due to the stage of retirement. In fact, sometimes the end of the work-life period weakens personal self-perception, as often there is a tendency to link personal fulfillment to successes obtained in the workplace. 

### 4.2. How ‘Active Ageing’ Is Perceived by the Respondents, and Its Domains 

Analyzing the insights deriving from interviews, it is possible to define three critical dimensions that compose, and favor, active aging from the point of view of older adults: (i) physical level; (ii) psycho-cognitive level; (iii) relational level. These three dimensions are closely linked to each other, all supported by the fundamental pillar of socialization.

Concerning the physical domain, bodily decay emerges for both target groups as an important factor: “If I will not be able to get up in the morning by myself… I prefer then not to live anymore” or “I do not want to live 130 years, but I want to live well”. Results from interviews showed a sense of fear towards the bodily changes that may occur in ageing. The reason for this is related to the fact that not only are these changes unpredictable, but physical issues may lead to the alteration of individual and social circumstances. This thought was shared by both analyzed groups. 

Sometimes the ageing process impacts older people’s lives to the point of leading them not to practice any physical activity or to be satisfied only with slight physical activities: “In this period I do not feel good, arthritis does not allow me to do activities”. Although the benefit of physical activity is largely recognized, older people often find it difficult to practice. Gyms are not usually appreciated by older adults, as they are clearly organized based on young people’s needs and they offer only strenuous physical activities. Participants from the older group consider gyms as a place not made for them and where they cannot be comfortable. On the other hand, many mature interviewees told the moderator that they practice Pilates and gentle gymnastics, not only to practice movement but also as a pretext to socialize. From this point of view, it is advisable to talk about ‘light movement’ or ‘physical activity’ to better identify the needs of the target group: “We do not only do the gym course… we drink a coffee together, but we also share moments in the dressing room… we have talk together and laugh and this is always something nice to do”.

Physical decay is strictly connected to the world of nutrition. However, while the mature target group is clearly convinced of the importance of a healthy diet to promote healthy aging, older adults are more likely to think that the relationship with food needs to be adapted to ageing. A healthy lifestyle is described by the mature target group in very different ways (variety of foods, fruits and vegetables every day, elimination of certain categories of foods such as sweets and red meat, slimming diets). For the older age group, it is important to “listen to your own body”, regarding food preferences and cravings: “I eat badly, for example before starting this interview I was munching on popcorn…”or “I am at that point in life that I can do whatever I want (…), not only negatives aspects of eating junk food should be considered”. This group of respondents focus more on the quality of raw materials and portions. Specifically, they tend to cook homemade dishes using seasonal products both from local shops and supermarkets (someone also mentioned buying organic food). In this regard, it is emphasized that food markets also allow socialization moments: “Unfortunately, many activities have been suspended because of COVID, including the town’s food market… going there was a weekly appointment”, for others “It is a moment in which I meet everyone”. 

In any case, no food needs or preferences arose spontaneously from respondents. In an implicit way, it is possible to identify emotional triggers via the concept of ‘dependence’. In fact, the future possibility of being dependent on someone else for personal care stimulates strong negative feelings. This could be the base from which to start encouraging healthier food habits and slight physical activity.

Concerning the psycho-cognitive level, there was no difference between the results emerging from the mature and the older group. Interviewees affirmed that they took care of their mental health, remaining active not only through common activities, such as reading, but also through practical activities (e.g., restoration, gardening). To look after mental health in later life, the two analyzed groups agree with the fact that, especially after retirement, people should invest time in engaging in their own interests and in new activities.

Summarizing, the needs of mature and older participants that emerged spontaneously in relation to the psycho-cognitive domain can be clustered as follows: (i) life-ling learning: people in late life would like to attend universities of the third age, they would appreciate sharing reading moments, attending language and technology courses. Interviewees stated that they want to keep up with modern society and today’s processes, and in order to do this they think that added value would be gained from participating in courses aimed at learning how to use a computer, the internet, WhatsApp, Facebook, Instagram, or at learning how to shop online to reduce physical fatigue; (ii) manual activities: this group includes various courses e.g., classes for cooking, knitting, painting, ceramics, restoration or gardening. Specifically, the aim in doing these courses is not only to learn, but also to ‘share moments’ and to ‘have goals’ and ‘take care of something’. According to the data which emerged, manual activities could allow mature and older people to be and to feel active both from the physical and cognitive perspective; (iii) volunteering for people, animals, or the environment (green care, cleaning, repairs). These activities emerged as significant factors in many interviews, providing a goal and a sense of usefulness. Especially, the older sample displayed the need to have opportunities to contrast with the common stereotype of older people’s purposelessness. In addition, participants perceive a sense of accomplishment in having and caring a pet. According to both the analyzed groups, this condition has several benefits, e.g., it is very supportive to as company to older adults living alone and it is also helpful in remaining active with physical activity; (iv) leisure: interviewees affirmed that they appreciate participating in exhibitions, visiting museums and theater, going to the cinema, and 9on organized trips. Regarding this last factor, it is pointed out that the older group is only willing to travel to destinations close to their own area, while the mature target group does not set limits to their travel plans. In fact, it is necessary to keep in mind that older adults are particularly attached to routine, belongings, and their environment [80] and they may find it difficult to stay far away from home for a long time. In this regard, the representation of an older adult as a person that needs assistance (in the simplest daily activities) clearly creates a sense of hostility among interviewees. In fact, some of them affirm: “If I have to become a vegetable that is not able to get out of bed, I prefer not to be here anymore” or “I do not want to think about how I will be”. 

In this context, knowledge and interest in co-housing solutions has been explored. What emerges is that these structures are still not known on a large scale, but for those that are familiar with them, co-housing is considered as a pleasant solution in case of need for assistance or in case of loneliness. On the other hand, nursing homes are negatively evaluated by participants, as they are associated with isolation and closeness to unknown people instead of loved ones. In fact, both mature and older interviewees highlighted the need to age at the side of important people in their life, which is not possible in residential facilities. Older adults showed a preference for hiring help at home (caregiver) as to them it appears as a better solution, even if it requires effort to find the “right” person. However, the mature sample was not able to express preferences about which solution they would prefer in case of assistance when older, but concerning nursing homes, mature participants affirmed that they and their own families might be reluctant to admit an ‘unknown’ person into the home. These findings are supported by the research of Palese et al. [82] and Santini et al. [83] because both studies reported that the ageing are not motivated to live in nursing homes. 

Overall, in order of preference, the solutions that are considered for assistance in late life are the following: (i) having help from children and staying at home is the most preferred solution, along with the desire not to weigh absolutely on their families, as previously pointed out by Halaweh et al. [52]; (ii) the possibility of hiring a person for day and night care such as a caregiver despite difficulty in accepting a ‘stranger’ at home; (iii) as a last option, admission to retirement houses, which as stated above have a bad reputation; some of the interviewees affirm: “Nursing homes look like camps”, “Nursing homes is where people wait for their own end” or “In this place you are together with other people but actually you are alone”.

The most accepted perspective is the less known scenario: co-housing for older adults. Although the appeal of these new housing solutions is high, strong doubts about their diffusion and about the economic methods of access are common. They are also anchored to an unreal imaginary, typical of the American continent, since some of them affirm: “But do they exist in Italy?” or “I imagine them as a movie house”, “I do not think they will be affordable”.

In line with different studies (e.g., [47,67]), the relational dimension is also an important aspect, as loneliness in this age group is common. The importance of the socio-relational domain is also highlighted by the fact that the interviewees enjoyed spending time participating in this study, having a chat with the moderator, expressing thoughts and opinions. Being in company and being listened to are widely appreciated, especially for older and mature adults after retirement. Older adults told the moderator that they would like to receive visits at home, perhaps also having the company of a dog or cat. One of them also suggested a solution that the municipality could adopt to help lonely people: “When separate collection of waste was introduced, the municipality sent young people around to explain the procedure… in this way they could employ volunteers who take turns door-to-door to visit older and lonely people”. However, relational initiatives appear to be more difficult to achieve in urban contexts than in the suburbs, where relationships are notoriously closer than in urban contexts. Some of the interviewees, indeed, living in the urban context affirmed: “We say goodbye in the morning and that’s it… even if we have a chat, it is something superficial” or “I do not know anything about my neighbors’ life, and they do not know anything about me… even if we live in the same building”. 

During interviews, the topic of transport was addressed, and it emerged as crucial for both the mature and the older sample. While most of the mature target still frequently drives, transport services emerge as very important especially for older adults who are not equipped to reach city centers, shopping centers and basic services (e.g., hospitals and post offices). In fact, not all older adults are able to move from home to city centers: “I am now paying a person who takes me shopping or running errands”. In this context, there are many initiatives that could be put in place to facilitate the use of public services by this demographic group, such as: (i) to introduce shelters and benches in all stops; (ii) to develop clear indications of waiting times through light panels; (iii) to introduce time slots dedicated to the over 60s to facilitate the use of vehicles (crowded in some time slots); (iv) to remove barriers inside the vehicles, as older interviewees told moderator that they had difficulty in getting in and out due to the height of steps and uncomfortable seats; (v) to organize dedicated shuttles that pick older people up at home and that take them to do their errands or visits (also favoring sociability); (vi) to increase the safety of cities, thus favoring evening or winter dark outings. Thus, these findings are in line with Pangbourne et al. (2018) [29], who stated that transport services need to be secure and easily usable, reliable and comfortable. 

## 5. Discussion and Implications

Although it is difficult to define a clear perspective of mature and older adults towards the late stages of life, it is important to encourage the planning of this life period at younger ages. Indeed, as widely recognized by many prior researchers, habits and experiences in earlier life directly affect later life’s quality. Hence, adopting a life course perspective, ‘formulas’ that lead to active and healthy aging should be encouraged in a more and more concrete way. To incentivize effective intervention programs in active ageing, Saito et al. [98] highlight that the initiatives need to be tailor-made based on specific needs of older people, targeting individuals with comparable experiences and preferences and exploiting existing community resources.

Data collected during interviews have allowed a deeper understanding of older adults, responding to the first research question “How do mature and older adults perceive the concept of ‘active ageing’? What are their current habits and expectations regarding health and social conditions?” (RQ1). Considering this first issue, results from the present study are in line with the large amount of literature highlighting the importance of promoting an active ageing. Thus, its inclusion in the political agenda of public institutions, governments, and private organizations is even more supported. However, a contrasting perspective on ageing is confirmed: while societies attach negative connotations to ageing [11], people directly involved in this process do not want to be affected by this belief and they have a positive and optimistic view of ageing. In addition, as was previously argued by Mathur and Moschis [30] and Moschis [31], qualitative research shows a picture of older adults with a dynamic lifestyle, full of passion and interests. This leads to increasing, selective and demanding requirements to satisfy their services’ needs. Therefore, the development of recreational and sports clubs for older adults, which allow the sharing of moments of leisure and pleasure, are certainly among the most interesting services to offer.

Findings also confirm what was reported by Rosseau [26], i.e., that older adults are particularly attentive to facilities they receive from governments and businesses. Specifically, these services concern the health and the socio-relational spheres. These domains see a strong interweaving between physical, psycho-emotional, and relational levels. Specifically, health and socio-relational dimensions are strictly connected by two pillars: independence and socialization. Concerning the health domain, it is necessary to consider that physical decay and cognitive health are the main important aspects that mature and older adults hope to preserve when ageing. Activities that allow spending time outdoors, hanging out with friends, meditating, taking care of the quality of sleep and feeding curiosity are certainly valid solutions to help this category of population to achieve their goals.

Despite this, older people’s habits are more interested in current well-being rather than in ensuring their future prosperity. In accordance with Silva and Fixina [66], older adults do not have a plan for the future; instead, an overall project for it should be prepared in advance. These elements find confirmation in the fact that interviewers declared that they practice physical activity, and they hang out at food markets not only to move, eat healthily and age well, but as a pretext to meet their peers and socialize. On the other hand, regarding the socio-relational perspective, this age group often feel lonely and vulnerable, especially in contexts influenced by retirement, distance from families and the COVID pandemic. In these cases, cohousing structures, i.e., housing projects characterized by strong social integration and based on mutual support, in which common spaces and services are shared, would be excellent solutions to be implemented locally, even in small, inhabited centers, perhaps grouping in a wider geographical area, to solve this problem. Moreover, in support of the literature, as the arising of physical issues related to ageing may be a barrier in having experiences outside of the home and in socializing, mature and older adults need to be included, with an active role in societies, communities, and activities. Insights collected during interviews confirmed previous research [49,52,67] highlighting that the most critical concerns of older people in addition to feeling lonely is depending on others and being a burden. For this reason, to promote active ageing by socially integrating mature and older adults, policy actions have to “support activities that foster social cohesion”, “provide access to life-long learning”, and “promote solidarity among the generations” [6] (p. 21). An example of a solidarity action that can be implemented is that of adopting a grandfather, to help sufficiently autonomous older people to still live in their own home, but with assistance and help from volunteers who every day undertake necessary tasks for the “grandparents” who have enrolled in this program.

In response to the second research question “*Which kind of health and social services are mainly requested and could be improved to support population towards active ageing?*” (RQ2), territorial services and private organizations can do a lot to trigger a continuous ‘exchange’, even beyond working age. Listening to the sample’s needs, many proposals for services and initiatives in favor of active ageing have emerged. As mature and older adults want to feel useful and socially active, public policies should emphasize the implementation of common spaces where they could learn something new, but also could apply themselves in having a goal and taking care of something. These sharing moments could involve activities of restoration, as well as cultivation of vegetable gardens. Meanwhile, they could imply the organization of specific events in which older adults share ideas, thanks to their knowledge and life experience. As an example, we remember the annual Aboriginal Storytelling Month, taking place in Canada, where older adults from across the province take part in activities reflecting their indigenous history and culture. In this context, co-housing solutions could offer great opportunities for older adults’ well-being and active ageing. However, as was also highlighted by Durante [90] and Sarlo et al. [92], in Italy co-housing initiatives and the knowledge of them have to increase. To achieve this, government could allocate funds to the implementation of these solutions, also incentivizing the diffusion of awareness among mature and older adults. 

In addition, as emerged from the interviews and in line with Rosseau [26], Mantovani et al. [77] and Tavares et al. [43], older adults find satisfaction in volunteering. For this reason, opportunities to engage them in activities aimed to help both others and the environment should be increased. These could involve municipal facilities such as repairs in playgrounds, activities which involve collaboration with schools, cleaning, and maintenance of greenery. 

Moreover, it is important to consider that not all mature and, especially, older adults are able to move independently, especially in the case of poor transport organization which characterizes rural areas. In these contexts, local government could encourage car sharing initiatives, to stimulate and facilitate movement outside of home. On the other hand, in addition to encouraging outings, some activities aimed at including this population could be organized door-to-door, giving them the possibility to socialize in their comfort zone. These initiatives could consist in visiting adults, especially the oldest, knitting, reading to them in case of absence of autonomy, and thus keeping them company. In doing so, pet therapy could play an important role and the adoption of a pet such as a dog, cat, rabbits, or birds may help those adults living alone not to feel lonely. In this regard, the development of ‘befriender services’, such as the creation of networks aimed at sharing time with older people by a telephone chat, could also be stimulating, establishing relationships that could be turned into home visits or outings.

## 6. Conclusions, Limits and Lines for Future Research

The ageing of the population is a characteristic of the modern demographic picture of many countries. The increase in life expectancy, the improvement of economic conditions, and technological and medical progress have changed the demographic structure of societies. This new demographic condition leads to the need for implementing possible interventions at social and territorial levels with the aim of increasing the well-being of the vulnerable older target citizens.

As emerged via the qualitative research, the concept of active ageing is perceived by mature and older adults in a positive and optimistic way (RQ1). The sample considered want to re-engage in life, continuing to be active, useful, and maintaining their self-esteem, social life, and independence. However, despite that older people’s major concerns are to preserve their physical abilities and social integration, this target group adopts behaviours focusing more on current well-being rather than future prospects. Thus, this highlights the need for incentivizing the importance of developing an ageing plan for mature and older adults. 

By assuming this perspective, the idea of later life as a stage characterised by dependency and isolation must cease to exist. The first step towards a radical evolution of the paradigm is to actively and deeply listen to older people’s needs. In fact, their well-being is strictly connected to this aspect, which often is lost from sight due to the tendency to focus on younger targets. 

Therefore, this context highlights the necessity to adapt person-centred integrated care guaranteeing access to an age-friendly environment and services, which engage mature and older adults with families and communities. The interviews conducted allowed investigation of the main barriers of public services in adults’ opinion and, starting from them, many proposals for service initiatives emerged (RQ2). Active ageing should be emphasized through activities which keep older adults both cognitive and physically active, such as learning on courses, restoration activities, or light movement initiatives. However, socialization and independence are increasingly important in the ageing process, and they have to be the core of overall interventions. Initiatives should be incentivized making it for older people to have access to them, such as facilitating transport to these places. In fact, this target group does not want to be a burden to families but, on the other hand, respondents reject the idea of moving to a retirement home. For this reason, local governments could encourage car sharing initiatives to promote older people’s independence, and nursing homes should be replaced by co-housing, which, even if well perceived by mature and older adults, is not as common a solution as it should be, at least in the Italian context under investigation.

This study offers support to the literature on active ageing but pays more attention to the older people’s perspective and needs, which often are neglected by previous studies. Nevertheless, the present research has some limitations. Due to the small sample size and the focus on a specific geographic area (central Italy), the findings cannot be generalised. Moreover, a single qualitative study does not allow a detailed view of older adults’ needs and perceptions towards active ageing. Thus, future research could be extended on a large scale, by adopting a quantitative approach, to improve the results’ generalizability as well as the external validity of the research. Additionally, cross-country studies may be realized to consider the impact of cultural factors on this target, and differences related to the cognitive status of the respondents (e.g., vedoval status, presence of a caregiver at home, use of technologies, etc.) could be taken into more consideration in future studies, to provide a better understanding of factors influencing the way older adults approach the later stage of their life.

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
