# Peer review of "Mature and Older Adults’ Perception of Active Ageing and the Need for Supporting Services: Insights from a Qualitative Study"

_ijerph, 2022, doi:10.3390/ijerph19137660_

Round 1

Reviewer 1 Report

It is an interesting paper and well organized. The authors classified the manuscripts in different parts: After the literature background on the concept and domains of active ageing with a focus on health and socio-relational dimensions regarding this issue. After that, they have put detail definitions on the methodology used; finally, they presents the results obtained from interviews followed five discusses the results and related implications and finished the main conclusions.  Conclusion is so long sometimes it is difficult to follow the interpretations of their results. May be they can simplify during the proof reading.

Author Response

RESPONSES TO COMMENTS OF REVIEWER 1

Comment 1:

“It is an interesting paper and well organized. The authors classified the manuscripts in different parts: After the literature background on the concept and domains of active ageing with a focus on health and socio-relational dimensions regarding this issue. After that, they have put detail definitions on the methodology used; finally, they presents the results obtained from interviews followed five discusses the results and related implications and finished the main conclusions”.

Response to Comment 1:

We thank a lot Reviewer 1 for this appreciation. We are glad that she/he found our paper well written and sufficiently clear with respect to its content and purposes.

Comment 2:

Conclusion is so long sometimes it is difficult to follow the interpretations of their results. May be they can simplify during the proof reading”.

Response to Comment 2:

Thank you for this careful suggestion. In the revision process, we made our best to shorten and simplify the Conclusion section to better highlight our research findings and theoretical contribution.

Reviewer 2 Report

The article uses a qualitative approach to investigate elders’ needs. The article reads very well and have coherent methods. Notwithstanding, I would like to share some comments and suggestions with the authors. Maybe the major point involves the rationale. It is not possible to find the rationale for studying people <60 years old over the article. The introduction and most of the literature reviewed involve older people. I suggest the authors to improve this point and correct the title as well. Please, provide a more detailed description of the methods, specially the characteristics of the participants. It would also be interesting to see more practical application and propositions.

Specific points:

Line 57 – please provide references. There are many good articles to cite instead of the WHO website.

Line 176 – please present this concept of active ageing in the beginning of the article. Many people, like me, might associate “active” with physical activity.

Author Response

Comment 1:

The article uses a qualitative approach to investigate elders’ needs. The article reads very well and have coherent methods. Notwithstanding, I would like to share some comments and suggestions with the authors”.

Response to Comment 1:

Thank you very much for appreciating our paper and for your precious recommendations aimed at improving its overall quality.

Comment 2:

Maybe the major point involves the rationale. It is not possible to find the rationale for studying people <60 years old over the article. The introduction and most of the literature reviewed involve older people. I suggest the authors to improve this point and correct the title as well”.

Response to Comment 2:

Thank you for your comment. We have decided to consider these two populations groups, since the literature divides the more mature population into two groups, namely "mature people" (50-60 years) (Gordon et al., 2002) and older adults (61 and more years) (Robbins et al., 2018). The choice of making a comparison between the two population groups is relevant as it allows to understand both the point of view of the subjects directly involved in active aging but also that of the subjects who will be involved in the coming years, in order to be able to develop current and innovative reference actions. This comparison covers a gap in the literature as there are no analyses that carry out a comparison of this kind. This has been better explained in the Introduction section and in the Methodology one and also the title has been reconsidered as well.

Comment 3:

Please, provide a more detailed description of the methods, specially the characteristics of the participants”.

Response to Comment 3:

Thank you very much for this further comment concerning the methodology. During the revision process, we improved the Methodology section, adding several information concerning the sample size (i.e. how we decided the sample size) and the questionnaire structure.

Unfortunately, we lack information on respondents’ demographic characteristics. As explained in the paper (see p. 9), data for this study were collected by Cias Innovation (belonging to Intertek Italia), which also contributed to the sample selection. Since the respondents were selected among the Intertek Italia’s community of panellists, they didn’t provide detailed information on them for privacy protection. Hence, we can inform just about the sample composition, but not its demographic characteristics. We are sure that you can understand our reasons.

Comment 4:

It would also be interesting to see more practical application and propositions”.

Response to Comment 4:

Thank you very much for this suggestion, we have enriched par. 5 (i.e., “Discussion and implications”) with some more practical applications and propositions as suggested.

Comment 5:

Specific points:

Line 57 – please provide references. There are many good articles to cite instead of the WHO website.

Line 176 – please present this concept of active ageing in the beginning of the article. Many people, like me, might associate “active” with physical activity”.

Response to Comment 5:

Thanks for these further suggestions. On p.2, we replaced the WHO website with the following references, published on top-ranked journals:

1) Elwood P, Galante J, Pickering J, Palmer S, Bayer A, Ben-Shlomo Y, et al. (2013) Healthy Lifestyles Reduce the Incidence of Chronic Diseases and Dementia: Evidence from the Caerphilly Cohort Study. PLoS ONE, 8(12): e81877;

2) Rus, V. A. (2019). The Role of Healthy Diet and Lifestyle in Preventing Chronic Diseases. Journal of Interdisciplinary Medicine, 4(2), 57-58.

Moreover, the concept of active ageing originally defined on p. 4 has been moved to the Introduction section, on p. 2, to make immediately clear to the reader the study domain.

Reviewer 3 Report

This is an interesting topic, and one that is of importance given the globally ageing population. Please find below a number of recommendations.

  1. I found this article difficult to read due to the English and grammatical errors. I have not commented on each of these specifically, and have only referenced the sentences where I struggled/was unable to understand the authors meaning:
    1. Lines 9 – 11 ‘The improvement in life expectancy, economic conditions, and technological and medical progresses have radically changed the demographic structure of societies, and, nowadays, ageing of population is characterising the modern demographic of many countries’. I am not sure what the authors mean by the second half of this sentence, and it may be better to make this into more than one sentence.
    2. Lines 23 – 25 ‘However, despite old people’s major concerns are to preserve their physical abilities and social integration, this target adopts behaviours focused more on the current well-being rather than in a future prospect.’ This sentence requires further clarity.
    3. Lines 64 – 66 ‘a lifestyle in an active ageing perspective is placed at the interplay between the individual and his/her surrounding, involving the biological, psychological and social domains of function.’ I am unsure the authors are trying to say in this sentence
    4. Lines 303 – 305 ‘On the other hand, diet individuals' responsibilities are to “maintain a normal body weight” along the life course and preferring a “diet high in fibre and low in animal fat and salt”.’ I am unsure that the authors are trying to say with this sentence.
    5. Lines 445 – 446 ‘(iv) perception and needs regarding the partner assistance sanitary’. I am unsure that the authors mean by this sentence.
  2. It is recommended that the authors re-consider their use of terminology when referring to older adults. As per the AMA Manual of Style (and also several journals which focus on research with older adults, including the Journal of American Geriatrics Society), words such as aged, elder, elderly, and seniors should not be used, as such terms connote discrimination or a stereotype. In particular, it is noted that the authors frequently refer to their research population as ‘elderly’ (which outside of this research is often used to describe frail individuals, and is seen in a negative way), and also as ‘old people’ and ‘seniors’. Additionally, research indicates that older adults do not like the term elderly when it is applied to them (the authors even note this in their own research findings). The AMA Manual of Style recommends using the following terms instead: older persons, older people, elderly patients, geriatric patients, older adults, older patients, aging adults, persons 65 years and older, or the older population. I would suggest the use of the term ‘older adults’ as this seems to be the most commonly used and accepted term.
  3. In the method section, the authors noted that 16 interviews were completed, and that there were selection criteria used to obtain the ‘mature’ and ‘senior’ samples. It would be useful to include a demographic table, so that it is clear to the reader how the ‘mature’ and ‘senior’ samples differed.
  4. In the results section, the differences between the two sample groups are rarely discussed. It would be helpful to at least include a statement to say ‘there was no difference’ if the groups did not differ on a particular topic.
  5. I note that 8-9 pages of the article are allocated to background literature, whereas only 3 ½ pages (or 4 ½ pages if you include the discussion) are allocated to the results of the research. I personally, would be more interested in seeing the results section fleshed out as feel that several of the topics have only been superficially touched on, rather than having a lengthy background and literature review.

Author Response

Comment 1:

This is an interesting topic, and one that is of importance given the globally ageing population. Please find below a number of recommendations”.

Response to Comment 1:

We thank a lot Reviewer 3 for the detailed and thoughtful suggestions, which inspired our revision in an attempt to improve the overall quality of the paper and make it more suitable for publication.

Comment 2:

I found this article difficult to read due to the English and grammatical errors. I have not commented on each of these specifically, and have only referenced the sentences where I struggled/was unable to understand the authors meaning:

    1. Lines 9 – 11 ‘The improvement in life expectancy, economic conditions, and technological and medical progresses have radically changed the demographic structure of societies, and, nowadays, ageing of population is characterising the modern demographic of many countries’. I am not sure what the authors mean by the second half of this sentence, and it may be better to make this into more than one sentence.
    2. Lines 23 – 25 ‘However, despite old people’s major concerns are to preserve their physical abilities and social integration, this target adopts behaviours focused more on the current well-being rather than in a future prospect.’ This sentence requires further clarity.
    3. Lines 64 – 66 ‘a lifestyle in an active ageing perspective is placed at the interplay between the individual and his/her surrounding, involving the biological, psychological and social domains of function.’ I am unsure the authors are trying to say in this sentence
    4. Lines 303 – 305 ‘On the other hand, diet individuals' responsibilities are to “maintain a normal body weight” along the life course and preferring a “diet high in fibre and low in animal fat and salt”.’ I am unsure that the authors are trying to say with this sentence.
    5. Lines 445 – 446 ‘(iv) perception and needs regarding the partner assistance sanitary’. I am unsure that the authors mean by this sentence.

Response to Comment 2:

Thank you very much for your precious suggestions. We have revised all the sentences considered, trying to better specify the meaning. Additionally, the paper has been revised by a native English speaker to improve its readability and remove grammatical errors.

Comment 3:

It is recommended that the authors re-consider their use of terminology when referring to older adults. As per the AMA Manual of Style (and also several journals which focus on research with older adults, including the Journal of American Geriatrics Society), words such as aged, elder, elderly, and seniors should not be used, as such terms connote discrimination or a stereotype. In particular, it is noted that the authors frequently refer to their research population as ‘elderly’ (which outside of this research is often used to describe frail individuals, and is seen in a negative way), and also as ‘old people’ and ‘seniors’. Additionally, research indicates that older adults do not like the term elderly when it is applied to them (the authors even note this in their own research findings). The AMA Manual of Style recommends using the following terms instead: older persons, older people, elderly patients, geriatric patients, older adults, older patients, aging adults, persons 65 years and older, or the older population. I would suggest the use of the term ‘older adults’ as this seems to be the most commonly used and accepted term”.

Response to Comment 3:

Thanks a lot for this recommendation. We agree on the fact that the use of some terms such as ‘elderly’, ‘old people’ and ‘senior’ could be misinterpreted by the scientific community. Therefore, following the reviewer’s suggestion, in line with the AMA Manual of Style, we avoided the use of the above terms and, in the current version of the paper, we always use “mature” and ‘older adults’ to refer to our research population.

Comment 4:

In the method section, the authors noted that 16 interviews were completed, and that there were selection criteria used to obtain the ‘mature’ and ‘senior’ samples. It would be useful to include a demographic table, so that it is clear to the reader how the ‘mature’ and ‘senior’ samples differed”.

Response to Comment 4:

Unfortunately, we lack information on respondents’ demographic characteristics. As explained in the paper (see p. 9), data for this study were collected by Cias Innovation (belonging to Intertek Italia), which also contributed to the sample selection. Since the respondents were selected among the Intertek Italia’s community of panellists, they didn’t provide detailed information on them, for privacy protection. Hence, we can inform just about the sample composition, but not its demographic characteristics. We could just discuss the difference in the result section, in an aggregate way .We are sure that you can understand our reasons.

Comment 5:

In the results section, the differences between the two sample groups are rarely discussed. It would be helpful to at least include a statement to say ‘there was no difference’ if the groups did not differ on a particular topic”.

Response to Comment 5:

Thank you for this precious recommendation. In the current version of the paper, we integrated more details wherever there were differences between the two sample groups. In the remaining parts, as you suggested, we specified that no differences were found.

Comment 6:

I note that 8-9 pages of the article are allocated to background literature, whereas only 3 ½ pages (or 4 ½ pages if you include the discussion) are allocated to the results of the research. I personally, would be more interested in seeing the results section fleshed out as feel that several of the topics have only been superficially touched on, rather than having a lengthy background and literature review”.

Response to Comment 6:

We appreciate a lot your recommendation to create a balance between the pages of the background literature and the results of the research. We developed the first version of the paper trying to focalize the results’ section on the most relevant points emerged, but we understand more details were needed to make the study more interesting. Thus, we integrated more insights as possible from data provided by the Cias Innovation.

Reviewer 4 Report

1.- It would be important to know.

The Demographics Variables: a.-Level of Instruction b.- Socio economic level of the participants.

2.- Before the interview. Did you evaluate the cognitive function of the participants?

3.- Did they have caregivers?:   

     a.- Informal?     b.- formal?.

4.- Some of them. Were Widows or widowers?

5.- How do you decide the Sample Size?

6.- You did not ask them, about new technologies.

Because, I think all these topics influence in the responses.

As you say, this research, does not have external Validity.

In order to give some recommendations about public policy, it´s necessary to do a quantitative study.

7.- I recommend to read this article:

a.- Rosa Ana Alonso Ruiz, Magdalena Saenz de Jubera Ocón, María Angeles Valdemoros San Emeterio and Ana Ponce de León Elizondo.

Digital Leisure: An opportunity for intergenerational Well being in Times of Pandemic.

Journal of New approaches in Educational Research.

2022 Vol 11 Nº1, 31-48

8.- And these others Notes.

a.- Aboriginal Storytelling Month an opportunity to share culture.

By Angela Brown 

Feb 1, 2022.

b.- Grandpa.

The Storyteller

Victor Issa is a figurative sculptor known for his narrative imagery.

This work is placed in the Storyteller´s area in the Lena Meijer Children´s Garden at Frederik Meijer Gardens & Sculpture Park. Michigan U.S.A.

Author Response

Comment 1:

1.- It would be important to know.

The Demographics Variables: a.-Level of Instruction b.- Socio economic level of the participants.

2.- Before the interview. Did you evaluate the cognitive function of the participants?

3.- Did they have caregivers?:   

     a.- Informal?     b.- formal?.

4.- Some of them. Were Widows or widowers?

5.- How do you decide the Sample Size?

6.- You did not ask them, about new technologies.

Because, I think all these topics influence in the responses”.

Response to Comment 1:

We thank Reviewer 4 for all the useful comments and suggestions. With specific regard to the Comment 1, we improved the Methodology section, adding several information concerning the sample size (i.e. how we decided the sample size) and the questionnaire structure.

Information concerned with the cognitive status of respondents were not considered in this study and we included this aspect among the limitation of the current research suggesting future explorations.

Unfortunately, we lack information on respondents’ demographic characteristics. As explained in the paper (see p. 9), data for this study were collected by Cias Innovation (belonging to Intertek Italia), which also contributed to the sample selection. Since the respondents were selected among the Intertek Italia’s community of panellists, they didn’t provide detailed information on them for privacy protection. Hence, we can inform just about the sample composition, but not its demographic characteristics. We are sure that you can understand our reasons.

Comment 2:

As you say, this research, does not have external Validity. In order to give some recommendations about public policy, it´s necessary to do a quantitative study”.

Response to Comment 2:

Thank you for this comment. Your suggestion has been exploited to improve the limitations of our study in the final part of the paper (p. 16).

Comment 3:

7.- I recommend to read this article:

a.- Rosa Ana Alonso Ruiz, Magdalena Saenz de Jubera Ocón, María Angeles Valdemoros San Emeterio and Ana Ponce de León Elizondo. Digital Leisure: An opportunity for intergenerational Well being in Times of Pandemic. Journal of New approaches in Educational Research. 2022 Vol 11 Nº1, 31-48

8.- And these others Notes.

a.- Aboriginal Storytelling Month an opportunity to share culture. By Angela Brown, Feb 1, 2022.

b.- Grandpa. The Storyteller Victor Issa is a figurative sculptor known for his narrative imagery.

This work is placed in the Storyteller´s area in the Lena Meijer Children´s Garden at Frederik Meijer Gardens & Sculpture Park. Michigan U.S.A.

Response to Comment 3:

We thank a lot Reviewer 4 for this comment. We read the article of Alonso Ruiz and colleagues (2022) and found it very interesting for our work. Thus we added this citation in our reference list and in text (on p. 7) to stress the contribution of new technologies to inter-generational well-being, in light of the current pandemic situation which has limited the opportunities of interaction between children and older adults.

As for the other notes, they further inspired our revision. Specifically, on p.14, we added the Aboriginal Storytelling Month as an example of event that is targeted to older adults with the purpose of stimulating knowledge and life-experience sharing.

Reviewer 5 Report

Dear Authors, 

your manuscript  discusses definitely interesting and relevant findings.  Demographic changes and aging trends  bring new challenges for societes. It is absolutely relevant to work on future strategies and interdisciplinary solutions involving seniors and their future needs, supportive in active aging. Although, the paper is generally well written and clearly structured, I suggest some improvement. The following point may be considered during revising this paper:

- it is worth to present the semi-structured questionnaire in methodology section in more clear way, I mean the certain questions not only mentioned topics.

Author Response

Comment 1:

Dear Authors, your manuscript discusses definitely interesting and relevant findings.  Demographic changes and aging trends  bring new challenges for societes. It is absolutely relevant to work on future strategies and interdisciplinary solutions involving seniors and their future needs, supportive in active aging. Although, the paper is generally well written and clearly structured, I suggest some improvement. The following point may be considered during revising this paper”:

Response to Comment 1:

Thank you very much for appreciating the relevance of our research topic and the overall quality of our paper.

Comment 2:

it is worth to present the semi-structured questionnaire in methodology section in more clear way, I mean the certain questions not only mentioned topics”.

Response to Comment 2:

We thank a lot for this recommendation. The methodology section has been improved accordingly, by enriching the description of the questionnaire (see p. 9). 

Round 2

Reviewer 2 Report

Thank you for addressing my comments.

Author Response

Thank you very much for appreciating our work.

Reviewer 3 Report

Thank-you for the revisions made to the paper, and for responding to the previous comments provided. Please find below feedback on the updated manuscript.

While there are still a number of grammatical errors in the paper (I have not commented on these as they will likely be picked up with final proofreading), this paper is now much easier to read and understand. Thank-you.  

My specific comments are:

-       - Lines 11-12 – I am still having difficulty understanding the updated sentence “Since ageing of population is characterizing the modern demographic of many countries, by adopting”. Are the authors trying to say “Since many counties now have ageing populations, by adopting ….”.

-        - Lines 48 – 49: “External variables linked to adult lifestyle - such as food and alcohol intake, smoking, social and economic condition - have a huge impact on people decline.” I am unsure what the authors are trying to say in the last part of this sentence ('have a huge impact on people decline').

-        - There are several occasions in the paper where the population is referred to as ‘old people’,  old adults, or old customers (rather than older people, older adults, or older customers). There are also several occurrences of the term 'elderly' still within the paper that the authors may wish to consider changing.

Author Response

Notes on the revision of the manuscript ID: ijerph-1700167

Title

MATURE AND ELDERLY PERCEPTION OF ACTIVE AGEING AND THE NEED FOR SUPPORTING SERVICES: INSIGHTS FROM A QUALITATIVE STUDY

Dear Editor and Reviewers,

We would like to thank you for the opportunity to revise again our paper to International Journal of Environmental Research and Public Health. We have tried to do our best to follow your precious suggestions.

Below, we addressed point by point the Reviewers’ comments.

------------

RESPONSES TO COMMENTS OF REVIEWER 3

Comment 1:

Thank-you for the revisions made to the paper, and for responding to the previous comments provided. Please find below feedback on the updated manuscript.

While there are still a number of grammatical errors in the paper (I have not commented on these as they will likely be picked up with final proofreading), this paper is now much easier to read and understand. Thank-you. 

Response to Comment 1:

Thank you very much for appreciating the effort made in improving the paper.

Comment 2:

Lines 11-12 – I am still having difficulty understanding the updated sentence “Since ageing of population is characterizing the modern demographic of many countries, by adopting”. Are the authors trying to say “Since many counties now have ageing populations, by adopting ….”.

Response to Comment 2:

Thank you for this comment, the sentence has been updated as you suggested.

Comment 3:

Lines 48 – 49: “External variables linked to adult lifestyle - such as food and alcohol intake, smoking, social and economic condition - have a huge impact on people decline.” I am unsure what the authors are trying to say in the last part of this sentence ('have a huge impact on people decline').

Response to Comment 3:

Thank you for this comment, the sentence has been revised according to Reviewer’s observation.

Comment 4:

There are several occasions in the paper where the population is referred to as ‘old people’, old adults, or old customers (rather than older people, older adults, or older customers). There are also several occurrences of the term 'elderly' still within the paper that the authors may wish to consider changing.

Response to Comment 4:

Thank you for this comment, we have uniformed the language in all the text by using always “mature” and “older adults”. We have also changed the term “elderly” accordingly to your suggestion.